# JND4135, a New Type II TRK Inhibitor, Overcomes TRK xDFG and Other Mutation Resistance In Vitro and In Vivo

**DOI:** 10.3390/molecules27196500

**Published:** 2022-10-01

**Authors:** Jie Wang, Yang Zhou, Xia Tang, Xiuwen Yu, Yongjin Wang, Shingpan Chan, Xiaojuan Song, Zhengchao Tu, Zhimin Zhang, Xiaoyun Lu, Zhang Zhang, Ke Ding

**Affiliations:** 1International Cooperative Laboratory of Traditional Chinese Medicine Modernization and Innovative Drug Development, Ministry of Education (MoE) of People’s Republic of China, College of Pharmacy, Jinan University, 601 Huangpu Avenue West, Guangzhou 510632, China; 2Guangzhou Institutes of Biomedicine and Health, Chinese Academy of Sciences, 190 Kaiyuan Avenue, Guangzhou Science Park, Guangzhou 510530, China; 3State Key Laboratory of Bioorganic and Natural Products Chemistry, Shanghai Institute of Organic Chemistry, Chinese Academy of Sciences, #345 Lingling Road, Shanghai 200032, China

**Keywords:** TRK, xDFG mutation, resistance, JND4135

## Abstract

The tropomyosin receptor kinases (TRKs) have been validated as effective targets in anticancer drug discovery. Two first-generation TRK inhibitors have been approved into market and displayed an encouraging therapeutic response in cancer patients harboring TRK fusion proteins. However, acquired resistance mediated by secondary TRK mutations especially in the xDFG motif remains an unsolved challenge in the clinic. Herein, we report the preclinical pharmacological results of JND4135, a new type II pan-TRK inhibitor, in overcoming TRK mutant resistance, including the xDFG mutations in vitro and in vivo. At a low nanomolar level, JND4135 displays a strong activity against wild-type TRKA/B/C and secondary mutations involving xDFG motif substitutions in kinase assays and cellular models; occupies the TRK proteins for an extended time; and has a slower dissociation rate than other TRK inhibitors. Moreover, by intraperitoneal injection, JND4135 exhibits tumor growth inhibition (TGI) of 81.0% at a dose of 40 mg/kg in BaF3-CD74-TRKA-G667C mice xenograft model. Therefore, JND4135 can be considered as a lead compound for drug discovery overcoming the resistance of TRK inhibitor drugs mediated by xDFG mutations.

## 1. Introduction

The tropomyosin receptor kinase (TRK) family includes three isoforms, TRKA, TRKB and TRKC (encoded by NTRK1, NTRK2, NTRK3 genes, respectively), whose primary binding ligands are the nerve growth factor (NGF), the brain-derived neurotrophic factor (BDNF)/neurotrophin-4 (NT4) and neurotrophin-3 (NT3), respectively [1,2]. Upon binding to their respective ligands, the TRK proteins dimerize, autophosphorylate and activate downstream signaling cascades such as the PLCγ-PKC, PI3K-AKT, and RAS-MAPK pathways, all of which play crucial roles in cell survival, growth, proliferation, and differentiation, respectively [3]. Abnormal activated TRK proteins including those with mutations, overexpression and fusion, have been detected and characterized as oncogenic drivers in a large variety of cancers including colorectal cancer, lung cancers, melanoma, acute myeloid leukemia, neuroblastoma, breast cancer and cutaneous cancers [4,5,6]. The collective identification of NTRK fusions in various human cancers stimulates the development of TRK inhibitors for cancer therapy [1,2,4,5].

Two first-generation TRK inhibitors, Larotrectinib (LOXO-101) [6] and Entrectinib (RXDX-101) [7], were approved by the U.S. Food and Drug Administration (FDA) in 2018 and 2019, respectively [8], for the treatment of TRK fusion-positive adult and pediatric cancers. Despite the initial encouraging therapeutic response to these first-generation TRK inhibitors in NTRK-fusion positive tumors regardless of patient age and tumor type, the duration of the response was invariably limited due to acquired resistance [9,10,11]. The amino acid mutations occurring at the solvent front (TRKA^G595R^, TRKB^G639R^, TRKC^G623R/E^), gatekeeper (TRKA^F589L^, TRKB^F633L^, TRKC^F617L^), ATP site (TRKA^V573M^, TRKB^V617M^, TRKC^V601M^) and xDFG motif (TRKA^G667S/C^, TRKB^G709C^, TRKC^G696A^) have been identified as the main acquired resistance mechanisms [12,13,14].

To overcome the acquired resistance to the first-generation TRK inhibitors, several second-generation TRK inhibitors such as Selitrectinib (LOXO-195, Bayer/Loxo Oncology) [10] and Repotrectinib (TPX-0005, Turning Point Therapeutics) [15] have been developed and progressed into different stages of clinical trials. It has been confirmed that Selitrectinib and Repotrectinib can overcome most of the mutant resistance, but their activities against TRK xDFG mutants are over 59.1-fold weaker compared with the wild type (WT). Several multikinase inhibitors such as Foretinib, Merestinib, Nintedanib, Ponatinib and Cabozantinib were also reported to preserve the sensitivity to mutant TRKs, including the xDFG mutant TRKA G667C, but their benefits to patients could be limited by potential off-target toxic and other side effects [16,17,18]. Other compounds such as IHMT-TRK-284 [19] have also been developed in an effort to overcome the TRK mutant resistance [20], but they are all in an early stage of development [21]. Consequently, xDFG mutation-induced drug resistance remains one of the major challenges for the development of second-generation TRK inhibitors.

Herein, we report the preclinical pharmacological results of JND4135, a type II pan-TRK inhibitor, which can overcome multiple resistant mutants in vitro and in vivo, including the xDFG mutants. JND4135 displays strong activity against wild-type TRKA/B/C and secondary mutants involving xDFG motif substitutions in kinase assays and cellular models. It binds to the TRK protein and occupies the TRK protein for an extended time, and shows 81.0% xenograft growth inhibition in animals when administered intraperitoneally. Further modification of JND4135 aimed at improving its oral bioavailability is in progress in our laboratory.

## 2. Results

### 2.1. JND4135 Is a Strong TRK Inhibitor against TRK WT and TRKA-G667C

Our group has previously reported a series of TRK inhibitors to treat NTRK fusion-driven cancers and neuroblastoma with over-expressed TRKB [22,23,24]. In order to develop new TRK inhibitors to overcome mutation resistance, we re-evaluated the TRA-G667C kinase inhibitory activities of those compounds and found that JND4135 has the potential to overcome xDGF mutant resistance. The kinase inhibitory activity of JND4135 and the four recognized TRK inhibitors, Entrectinib, Larotrectinib, Selitrectinib and Repotrectinib against TRKA, TRKB, TRKC and TRKA-G667C (IC_50_) (Figure 1), respectively, were evaluated using the established FRET-based Z’-Lyte assay. We found that JND4135 potently inhibits TRKA, TRKB, TRKC with IC_50_ values of 2.79 ± 1.17, 3.19 ± 1.76 and 3.01 ± 0.43 nM, respectively, which is similar to the potency of Entrectinib (2.25 ± 0.58, 2.89 ± 1.44 and 2.26 ± 0.41), Larotrectinib (3.20 ± 0.50, 5.22 ± 1.87 and 2.82 ± 1.22), Selitrectinib (2.59 ± 0.44, 3.58 ± 1.40, 1.87 ± 0.54) and Repotrectinib (1.85 ± 0.41, 2.85 ± 1.82 and1.75 ± 0.25) (Figure 1B–D, Appendix A). However, all four reported TRK inhibitors Entrectinib, Larotrectinib, Selitrectinib and Repotrectinib displayed approximately 20 to 400-fold weaker activities against TRKA-G667C mutant kinase than TRKA-WT with IC_50_ values of 40.35 ± 5.27, 1196 ± 41.07, 287.3 ± 17.96, and 52.21 ± 9.59 nM, respectively. Interestingly, JND4135 retains even stronger kinase inhibitory activity against TRKA-G667C than TRKA-WT with an IC_50_ value of 0.83 ± 0.06 nM. These data imply that JND4135 has a potential to overcome TRK xDFG mutant resistance (Figure 1E, Appendix A).

### 2.2. JND4135 Overcomes TRK xDFG and Other Mutant Resistance in BaF3 Stable Models

To investigate the anti-proliferation effect of JND4135 and reported TRK inhibitors in cells, we constructed a panel of BaF3 cells stably expressing wild-type TRK fusion proteins (CD74–TRKA, MPRIP-TRKA, ETV6-TRKB, QKI-TRKB, ETV6-TRKC and EML4-TRKC) and corresponding resistant mutant TRK fusion proteins including xDFG (TRKA^G667C^, TRKB^G709C^, TRKC^G696C^), solvent front (TRKA^G595R^, TRKB^G639R^, TRKC^G623R^), gatekeeper (TRKA^F589L^, TRKB^F663L^, TRKC^F617L^) and ATP sites (TRKA^V573M^, TRKB^V617M^, TRKC^V601M^). In BaF3 cells harboring wild-type TRKA/B/C, JND4135 displays strong activities with IC_50_ values of 2.4–23.1 nM regardless of 5′ fusion partners, which are similar to those of both first generation TRK inhibitors (Larotrectinib and Entrectinib) and second generation TRK inhibitors (Selitrectinib and Repotrectinib) (Table 1). In cells with solvent front mutations, JND4135 and 2nd generation TRK inhibitors (Selitrectinib and Repotrectinib) retain the potency with IC_50_ values ranging from 9.4 to 33.7 nM, but 1st generation TRK inhibitors (Larotrectinib and Entrectinib) display more than 66-fold reduced activities. For gatekeeper mutations, JND4135 maintained a moderate activity (37.3–178 nM), while Larotrectinib showed a more than 45-fold decrease in activity (>423 nM). For ATP site mutation such as TRKA^V573M^, TRKB^V617M^, and TRKC^V601M^, JND4135 displays stronger activity with IC_50_ values of 9.0 ± 4.8, 3.9 ± 2.6 and 11.8 ± 5.7 nM, respectively, than Entrectinib (>101 nM), Larotrectinib (>609 nM), Selitrectinib (>26.2 nM) and Repotrectinib (>29.4 nM) (Table 1). For xDFG mutation such as TRKA^G667C^, TRKB^G709C^ and TRKC^G696C^, JND4135 displays potent activity with IC_50_ values of 1.4 ± 0.2, 18.7 ± 2.9 and 1.3 ± 0.7 nM, respectively, which are approximately 15–95, 87–592, 9–76 and 5–33 fold better than Entrectinib, Larotrectinib, Selitrectinib and Repotrectinib (Table 1).

All these results show that JND4135 has an excellent anti-proliferative activity in cell models harboring WT TRK and resistant mutants, and has strong advantages over the first and second generation of TRK inhibitors especially against the xDFG and ATP site mutant resistance.

### 2.3. JND4135 Dose-Dependently Inhibits the Phosphorylation of TRKs WT and xDFG Mutation and Its Downstream Signal Molecules

In order to demonstrate the inhibitory effect of JND4135 on TRK phosphorylation, we performed western blot analysis in BaF3-CD74-TRKA-WT, BaF3-CD74-TRKA-G595R, BaF3-CD74-TRKA-G667C cells and TRKB/C homologous mutant cells after the treatment of JND4135 for 6 h. JND4135 displayed a dose-dependent inhibition of TRK signal pathway in all cells (Figure 2). In BaF3-CD74-TRKA models, JND4135 decreased the phosphorylation levels of TRKA and its downstream signaling proteins ERK, PLCγ, from 8 nM for TRKA-WT, 40 nM for TRKA-G595R and 1.6 nM for TRKA-G667C cells (Figure 2). We also observed that JND4135 suppressed the TRKB/C signal pathway at a similar dose in BaF3-ETV6-TRKB and BaF3-ETV6-TRKC models (Figure 2B,C). In parallel experiments, Larotrectinib and Selitrectinib showed limited inhibition of the activation of TRKs-xDGF mutants such as TRKA-G667C, TRKB-G709C and TRKC-G696C even at the concentration of 200 nM, but JND4135 could effectively inhibit the activation of TRKs-xDGF mutants at a dose of 8.0 nM (Appendix A). We also investigated the effects of JND4135 on cellular signal pathway alterations in parental BaF3 cells (Appendix A), which were lack of TRK fusion proteins and driven by interleukin-3 (IL-3). It was shown that JND4135 treatment increased the phospho-ERK levels, which probably resulted from negative feedback after JND4135 treatment in parental BaF3 cells.

Interestingly, it was noted that TRK total protein decreased at high concentration JND4135 treatment in some BaF3-TRKs cells, especially the TRKC fusion proteins (Figure 2 and Appendix A), which were consistent with reported data of TRK inhibitors by some other groups [17,25,26,27].

### 2.4. JND4135 Induces G0/G1 Phase Arrest and Apoptosis in BaF3–CD74-TRKA-G667C Cells

To better understand the distinctive activity of JND4135 against TRKA G667C, cell cycle and apoptosis analysis were performed by flow cytometry in BaF3–CD74-TRKA-G667C cells. JND4135 could induce perceptible G0/G1 phase arrest and apoptosis in a dose-dependent manner in BaF3–CD74-TRKA-G667C cells. In contrast, little effect of Larotrectinib and Selitrectinb were observed on cell cycle or apoptosis in BaF3-CD74-TRKA-G667C cells even at a concentration of 50 nM (Figure 3A,B). Western blot analysis further validated that JND4135 dose-dependently diminished the cell cycle related protein levels of cyclin-dependent kinase 2 (CDK2), Cyclin D2, and activated the caspase apoptotic cascade resulting in the cleavage of PAPR in BaF3–CD74-TRKA-G667C cells (Figure 3C).

### 2.5. Prolonged Occupation of JND4135 in TRKs Protein

Inspired by the activities against kinases and in cells, we used bio-layer interferometry (BLI) to investigate the binding affinity of JND4135 on TRKC and TRKC-G696C and elucidate its mechanism. These results are consistent with the kinase inhibitory activities (Appendix A). Our results showed that the binding affinity of JND4135 to the TRKC-WT with K_d_ of 2.57 nM, which is similar to that of Entrectinib (3.47 nM), Larotrectinib (5.38 nM), Selitrectinib (1.74 nM) and Repotrectinib (1.74 nM) (Appendix A, Figure 4A). However, JND4135 displays stronger binding affinity to TRKC-G696C protein with K_d_ = 3.68 nM, which is approximately 7.11, 10.27, 4.05 and 9.51-fold better than Entrectinib (26.2 nM), Larotrectinib (37.8 nM), Selitrectinib (14.9 nM) and Repotrectinib (35 nM). Interestingly, we also found that the dissociation rates of JND4135 with TRKC-WT and TRKC-G696C were 2.78 × 10^−4^ 1/s and 1.35 × 10^−3^ 1/s, respectively, which were approximately 3.81–27.85 fold slower than that of control drugs (Appendix A, Figure 4B). Therefore, once binding with TRKs protein, JND4135 could occupy the protein for a longer time than the control drugs.

### 2.6. JND4135 Is a Type II TRK Inhibitor

The crystal structure of JND4135 in a complex with TRKC was determined in our previous studies [22]. An in silico study showed that JND4135 can also be successfully docked into the mutant forms (TRKA^G595R^, TRKA^G667C^, and TRKAV^573M^) in a type II binding mode (Figure 5). The amide group on JND4135 forms a hydrogen bond interaction with the backbone nitrogen of D668 in the protein and its (isopropyl) benzyl moiety is bound deep in the hydrophobic region formed around the DFG-out domain. To validate the binding mode and further analyze the interactions between JND4135 and mutant proteins, the docking structures were submitted to lengthy (500 ns) molecular dynamic simulations. During the simulations, JND4135 remained stable in the binding site as indicated by the root mean square deviation (Appendix A). Although the residues in TRKA^G667C^ and TRKA^V573M^ are larger, JND4135 has an alkyne at this position and there is little steric conflict affecting the binding mode of the compound, compared to those in wild-type protein (Figure 5). The 4-aniline moiety in JND4135 extends into the polar groove formed by Arg599 and Asp596 in wild-type TRKA, TRKA^G667C^, and TRKA^V573M^. In TRKA^G595R^, Arg595 replaces Arg599 and forms a hydrogen bond interaction with Asp596. The 4-aniline moiety in JND4135 can still twist and interact with this arginine, thereby retaining its activity.

### 2.7. In Silico Molecular Mechanism Supporting the Long Occupation Time of JND4135

Infrequent metadynamics (InMetaD) has been established and successfully applied in multiple studies of unbinding kinetics [28,29,30]. Using this method, the unbinding process with occupation times of up to several seconds can be revealed in atomistic detail. We investigated the unbinding mechanisms of JND4135 and Larotrectinib from TRKA^G667C^ for comparison. For each system, ten independent InMetaD simulations were performed as in previous studies [28]. The individual unbinding time from each simulation was rescaled to calculate the occupation times for JND4135 and Larotrectinib, which were 266.0 s and 7.5 s, respectively. The calculated occupation times were in good agreement with the experimental observations. From the unbinding trajectories, we found that both JND4135 and Larotrectinib dissociated from the “allosteric channel”. JND4135 binds to TRKA^G667C^ with the DFG-out conformation, in which the activation loop occupies the unbinding channel. The methylphenyl moiety was held by Met671 and Phe521 of the protein, hindering the dissociation of JND4135. During the unbinding process, an intermediate state was observed when the imidazo [1,2-a] pyrazine moiety was trapped in the channel, which might contribute to the long occupation time of JND4135 (Figure 6 and Appendix A). In all of the unbinding trajectories, it was found that Larotrectinib bound to the DFG-in form, the activation loop was in the open form and Larotrectinib dissociated rapidly. In this conformation, the inhibitor can directly dissociate through the “allosteric channel” without being hindered (Figure 6 and Appendix A.

### 2.8. JND4135 Suppresses Tumor Growth in BaF3–CD74-TRKA-G667C Xenograft Model

Considering the overall promising results of JND4135 in vitro, the antitumor efficacy of JND4135 in vivo was performed in a BaF3–CD74-TRKA-G667C xenograft model. Mice were intraperitoneally administrated with vehicle or JND4135 once daily (20 and 40 mg/kg/day, respectively) for 12 consecutive days. JND4135 was well tolerated in both tested groups with no mortality or significant body weight loss during the whole in vivo experiment (Figure 7A). As shown in Figure 7B, JND4135 dose-dependently inhibited the tumor progression with TGI (tumor growth inhibition) of 34.2% and 81.0%, at a dose of 20 and 40 mg/kg, respectively. At the end of the treatment, tumor tissues were collected and lysed for WB detection. The results showed that JND4135 could effectively inhibit the activation of TRKA and downstream molecules such as PLCγ-1 and ERK. We also found that cleavage-activated Caspase-9 increased in tumor tissues treated with JND4135 at a dose of 20 and 40 mg/kg.

## 3. Discussion

The tropomyosin receptor kinase (TRKA/B/C) has been validated as an effective target for anticancer drug discovery, but due to secondary TRK mutations especially in the xDFG motif, mediated cancer resistance is still an unmet clinical need. It was previously reported that TRK xDFG substitutions such as TRKA G667C and TRKC G696C enhanced the steric hindrance and stabilized the inactive (DFG-out) conformations of the kinases, which could allow these kinases to be resistant to type I TRK inhibitors. Consistently, type II inhibitors have the potential to overcome xDFG-mutant resistance [17,18].

Although several second TRK inhibitors such as Selitrectinib have displayed the ability to overcome kinase mutant resistance, they are all type I inhibitors and more sensitive to solvent-front than xDFG mutations [18]. A few multi-kinase inhibitors such as ponatinib and cabozantinib show good activity against TRKA^G667C^ mutant, but weak activity against wild-type TRKA/B/C or solvent-front mutations (Appendix A). Consequently, we developed and identified JND4135 as a type II TRK inhibitor, with a broad-spectrum anti-TRK activity in cells harboring widetype and mutations in solvent front, ATP sites, gatekeeper and xDFG (Table 1). Our results strongly imply that the TRK inhibitory activity of JND4135 is significantly different from that of other reported type I and type II inhibitors.

It has been reported that type II multi-kinase inhibitors such as cabozantinib and foretinib have higher affinity to TRKA xDFG mutants than type I TRK selective inhibitors, which possibly contributes to the sensitivity to xDFG mutant resistance [17,18]. In agreement with reported data, we found JND4135 possesses a lower binding constant (Kd) against TRKC-G696C than Larotrectinib, Entrectinib, Selitrectinib and Repotrectinib. JND4135 can preserve kinase inhibitory activity against TRKA-G667C with an IC_50_ value of 0.83 ± 0.06 nM, and is 3.3-fold stronger than TRKA-WT. Furthermore, we found that JND4135 possesses a longer occupation time in TRKs protein than Entrectinib, Larotrectinib, Selitrectinib and Repotrectinib. These unique properties of JND4135 could possibly maintain its efficacy for a longer time.

Based on in vivo and in vitro studies, we confirmed that JND4135 is an excellent drug lead, which not only possesses a broad antitumor spectrum, but also exhibits different target affinity kinetics from previously reported molecules. Intraperitoneal administration of JND4135 in mice shows a potent activity and could overcome Selitrectinib-refractory TRK xDFG mutant resistance. However, JND4135 displayed almost no absorption via oral administration in rats (Appendix A). The modification of the structure of JND4135 is ongoing to obtain an oral bioavailable drug candidate.

In summary, we report the preclinical pharmacology results of JND4135, a type II pan-TRK inhibitor, in overcoming TRK mutant resistance including xDFG mutant in vitro and in vivo. JND4135 displays strong activity against wild-type TRKA/B/C and secondary mutants involving substitutions of the xDFG motif in cells, possesses an extended occupation of TRK protein with a slower binding and dissociation, and shows 81.0% xenograft growth inhibition in animals. Further modification of JND4135 to enhance its oral bioavailability is in progress in our laboratory.

## 4. Materials and Methods

### 4.1. Cell Lines

Murine BaF3 cells were purchased from JRCB cell bank. The BaF3 cell lines, which stably express natively-fused TRKA/B/C or various point mutants, were self-established by following previously described procedures. Briefly, NTRK genes were totally synthesized and cloned into pCDNA3.1(+) vectors (Invitrogen, Carlsbad, CA, USA). The pCDNA3.1(+) plasmids containing different NTRK-fusion genes were introduced into BaF3 cells by electroporation (Amaxa Nucleofector II device) using the “Amaxa Cell Line Nucleofector Kit V” (VCA-1003) according to the manufacturer’s directions. Stable strains were established by selection of cells 2 weeks post-transfection with 800 μg/mL G418 (Merck, Whitehouse Station, NJ) and further selection after the withdrawal of interleukin-3 (IL-3, R&D) for 2 weeks. All of BaF3 stable cell lines were verified by DNA sequencing, protein expression and anti-proliferation activity against positive drugs. Parental BaF3 cells were cultured in RPMI 1640 media (Biological Industries) supplemented with 10% fetal bovine serum (Biological Industries), 10 ng/mL IL-3 and 1% penicillin-streptomycin solution (Biological Industries), while all NTRK fusion-transformed BaF3 stable cell lines were cultured in the same medium in the absence of IL-3. All cells were grown at 37 °C in a humidified 5% CO_2_ atmosphere.

### 4.2. Agents

3-((6-(4-aminophenyl)imidazo[1,2-a]pyrazin-3-yl)ethynyl)-N-(3-isopropyl-5-((4-methylpiperazin-1-yl)methyl)phenyl)-2-methylbenzamide, JND4135, was designed and synthesized in our laboratory. Entrectinib (RXDX-101), Larotrectinib (LOXO-101), Selitrectinib (LOXO-195) and Repotrectinib (TPX-0005) were purchased from the Selleckchem Company (Houston, TX, USA). These compounds were dissolved in DMSO (Sigma-Aldrich, St. Louis, MO, USA) at a concentration of 10 mmol/L and the solution was stored at −20 °C. Primary antibodies against TRK(pan) (92991S), PLCγ1 (5690S), ERK1/2 (4695S), p-TRKA (Y674/675)/TRKB (Y706/707)(4621S), p-ERK1/2 (T202/Y204)(4370S), p-PLCγ1 (Y783)(14008S), Cyclin D2 (3741S), CDK2 (2546S), Caspase-3 (14220S), Caspase-9 (9508S), PARP (9532S), GAPDH (2118S) and anti-rabbit (7074S) or anti-mouse (7076S) IgG HRP-linked secondary antibodies were purchased from Cell Signaling Technology (Boston, MA, USA). Primary antibodies against phospho-TRKA/TRKB/TRKC (ab197071) and CDK4 (ab199728) were purchased from Abcam.

### 4.3. Anti-Proliferation Assay In Vitro

Cells were placed in 96-well plates (5000~8000/well) in complete medium. After incubation overnight, the cells were exposed to various concentrations (0.000038~10 µM) of JND4135 for a further 72 h. Cell proliferation was evaluated by a Cell Counting Kit 8 (CCK8, CK04, Dojindo Laboratories, Kumamoto, Japan). IC_50_ values were calculated by concentration-response curve fitting using GraphPad Prism 5.0 software. Each IC_50_ value is expressed as mean ± SD.

### 4.4. In Vitro Enzymatic Activity Assay

TRKA/B/C and the Z′-Lyte Kinase Assay Kit were purchased from Invitrogen. Assays were performed according to the manufacturer’s instructions. The concentrations of kinases were determined by optimization experiments and the concentrations used were 1.22 ng/μL for TRKA, 0.056 ng/μL for TRKB and 1.73 ng/μL for TRKC. The solutions of the compounds were diluted from 10^−10^ M to 1 × 10^−4^ M in DMSO and a 400 μM solution of the compound was prepared with 4 μL of the solution of the compound dissolved in 96 μL water. Then, a 100 μM ATP solution in 1.33× kinase buffer was prepared and a kinase/peptide mixture containing 2× kinase and 4 μM Tyr1 peptide (PV3190; Invitrogen) was prepared immediately prior to its use. A kinase/peptide mixture was prepared by diluting Z’-LYTE Tyr1 peptide (PV3190; Invitrogen) and kinase in 1× kinase buffer, and an 0.2 μM Tyr1 phosphopeptide solution was made by adding Z’-LYTE Tyr1 phosphopeptide to 1× kinase buffer. The final 10 μL reaction solution consists of 12.2 ng TRKA/0.56ngTRKB/17.3ngTRKC, 2 μM Tyr1 peptide in 1× kinase buffer. For each assay, 10 μL of a kinase solution, including 2.5 μL of the compound solution, 5 μL of the kinase/peptide mixture, and 2.5 μL of the ATP solution was prepared. The plate wells were mixed thoroughly and incubated for 1 h at room temperature (rt). Then 5 μL of the development solution was added to each well and the plate was incubated for 1 h at rt; the phosphopeptides were cleaved in this time. Finally, 5 μL of stop reagent was added to stop the reaction. For the control setting, 5 μL of the phosphopeptide solution instead of the kinase/peptide mixture was used as a 100% phosphorylation control. The 1.33× kinase buffer (2.5 μL) was used instead of ATP solution as a 100% inhibition control, and 2.5 μL 4% DMSO instead of compound solution was used as the 0% inhibitor control. The plate was measured on an EnVision Multilabel Reader (Perkin-Elmer). Curve fitting and data presentation was performed using GraphPad Prism, version 5.0. Every experiment was repeated at least 2 times.

### 4.5. Analysis of Cell Cycle and Cell Apoptosis by Flow Cytometry

For analysis of the cell cycle, cells were harvested and washed twice with PBS after treatment for 24 h with indicated drugs or DMSO. Approximately 6 × 10^5^ cells were resuspended in 150 μL BD Cytofix/Cytoperm buffer solution (#554722, BD). After 20 min at 4 °C, the cells were washed twice with BD perm/wash buffer (#554723, BD) and incubated with 200 μL of dyeing buffer (containing 0.1 mg/mL propidium iodide, 2 mg/mL RNase A) in the dark for 20 min at 4 °C. The cells were then analyzed on a Guava easyCyte flow cytometer (Merck, Whitehouse Station, NJ, USA). For analysis of cell apoptosis, cells were harvested and washed twice with ice-cold PBS after treatment for 48 h with indicated drugs or DMSO. Approximately 6 × 10^5^ cells were resuspended in 100 μL 1 × BD binding buffer solution (#556454, BD), and then incubated with Annexin V-PE (#556422, BD) and 7-ADD (#559925, BD) in the dark for 15 min. Finally, 400 μL of a 1 × BD binding buffer solution was replenished. The cells were then analyzed on a Guava easyCyte flow cytometer (Merck, Whitehouse Station, NJ, USA).

### 4.6. Molecular Docking

The structures of the TRKA (PDB: 6PL1) was obtained from the Protein Data Bank (http://www.rcsb.org/pdb, accessed on 8 April 2021) and the mutant forms (G595R, G667C, and V573M) were prepared and modeled by the Protein Preparation Wizard (Schrödinger, LLC, New York, NY, USA, 2019) using Prime. JND4135 was prepared using Ligprep, and induced fit docking was used for the molecular docking with the pose of JND4135 in TRKC as the reference (PDB: 6KZD). Residues within 5 Å of any ligand pose were refined and Glide XP was used to score the docking poses.

### 4.7. Molecular Dynamic Simulations

The poses from docking were subjected to molecular dynamics simulations using GROMACS 2020.4 [31]. The AMBER ff99SB-ILDN force field [32] was used for the protein and the general AMBER force field [33] was used for the ligands. The restrained electrostatic potential-derived charges were used for the ligands with the electrostatic potential calculated at HF/6–31G* by Gaussian 09 [34]. The TIP3P solvent model was used to solvate the system. A time step of 2 fs was used in all of the simulations. LINCS algorithm and particle mesh Ewald method were used. The system was relaxed before simulation, and simulated in the NPT ensemble at 300 K and a pressure of 1 atm. Production MD simulations were carried out for 500 ns with conformations stored every 100 ps.

### 4.8. Infrequent Metadynamics Simulations

Infrequent metadynamics (InMetaD) was used with Plumed 2.6 [35] patched GROMACS 2020.4. In line with previous studies [28], two collective variables (CVs) were used in InMetaD. In brief, CV1 was defined as the distance between ligand and the binding pocket defined by the center of mass of the ligand heavy atoms and the center of mass of the Cα of residues within 5 Å of the ligand. CV2 was defined as the the number of water molecules (represented by their oxygen atoms) within 8 Å of the ligand center of mass. The initial Gaussian height was 0.2 kcal/mol. The pace for the bias deposition was 10 ps. The obtained empirical cumulative distribution function (ECDF) fitted well with the theoretical cumulative distribution function (TCDF) for a Poisson process. The *p*-values from the Kolmogorov−Smirnov (KS) test were 0.58 and 0.48, respectively.

### 4.9. Western Blot Analysis

Cells were treated with various concentrations of each tested compound for a designated time. Then cells were lysed using 1 × SDS sample lysis buffer (CST recommended) with protease and phosphatase inhibitors. Cell lysates were loaded and electrophoresed onto 8–12% SDS-PAGE gel, and the separated proteins were then transferred to a PVDF film. The films were blocked with 5% fat-free milk in TBS solution containing 0.5% Tween-20 for 4 h at room temperature, then incubated with the corresponding primary antibody (1:1000–1:200) overnight at 4 °C. After washing with TBST, HRP-conjugated secondary antibody was incubated for 2 h. The protein signals were visualized by ECL Western Blotting Detection Kit (Thermo Scientific, Waltham, MA, USA), and detected with Amersham Imager 600 system (GE, Boston, MA, USA).

### 4.10. Octet Binding Assay

Biolayer interferometry (BLI) assays were performed with the Octet K2 machine (ForteBio, Inc., Fremont, CA, USA). Human TRKC WT and TRKC^G696C^ proteins (residues 526–839 from the kinase domain) were expressed and purified as described previously [28]. The purified TRKC or TRK^G696C^ protein (100–120 μg) was biotinylated (EZ-Link^®^ NHS-Biotin Reagents, Cat#21343, Thermo) by 5:1, and then incubated for 1 h at rt. Excess biotin was removed by column chromatography (PD MiniTrap^TM^ G-25 Desalting column, Lot#17030445), and the residue was diluted to 400 μL with PBS and immobilized onto Super streptavidin biosensor chips (ForteBio, Cat#18–5057). A duplicate set of sensors was incubated in an assay buffer (PBS containing 0.02% Tween20 and 0.1% BSA) with 5% DMSO but without protein for use as a background binding control. Both sets of sensors were blocked with a solution of 10 μg/mL biocytin for 1 min at rt. Drug concentrations ranged from 0.74 to 180 nmol/L (6 doses). Data was analyzed by the FortéBio Data Analysis 8.1 software (FortéBio, Fremont, CA, USA) using a double reference subtraction (sample and sensor references).

### 4.11. In Vivo Experiments

Female CB17-SCID mice (6–8 weeks of age) were purchased from Vital River Laboratory Animal Technology Inc. (Beijing, China). All animal experiments were carried out under protocols approved by the Institutional Animal Care and Use Committee of the Medical College of Jinan University. An amount of 2 × 10^6^ BaF3-CD74-TRKA-G667C cells were injected subcutaneously in the right flank of the SCID mice. When the mean tumor volume reached 100–200 mm^3^, mice were randomly grouped based on the tumor volume. JND4135 was dissolved with the solution containing 2% DMSO, 20% Cremophor, 8% absolute alcohol and 70% normal saline. The animals were treated with JND4135 for 12 consecutive days once daily by oral gavage with 20 mg/Kg (*n* = 6) or 40 mg/Kg (*n* = 6) of JND4135 and vehicle (*n* = 6), respectively. Tumor volume and body weight were monitored once every two days. Tumor volume was calculated as L×W^2^/2, where L is the length and W the width of the tumor. Tumor volumes were compared using one-way ANOVA with post hoc intergroup comparison using the Tukey test.

### 4.12. Statistical Analysis

Data are expressed as the mean ± SD of three independent experiments. Comparisons between two groups used a two-tailed Student’s *t* test, and comparisons among multiple groups involved one-way ANOVA with post hoc intergroup comparison using the Tukey test. Differences with *p* < 0.05 and *p* < 0.01 were considered as significant or very significant, and marked as * and **, respectively.

## Figures and Tables

**Figure 1 molecules-27-06500-f001:**
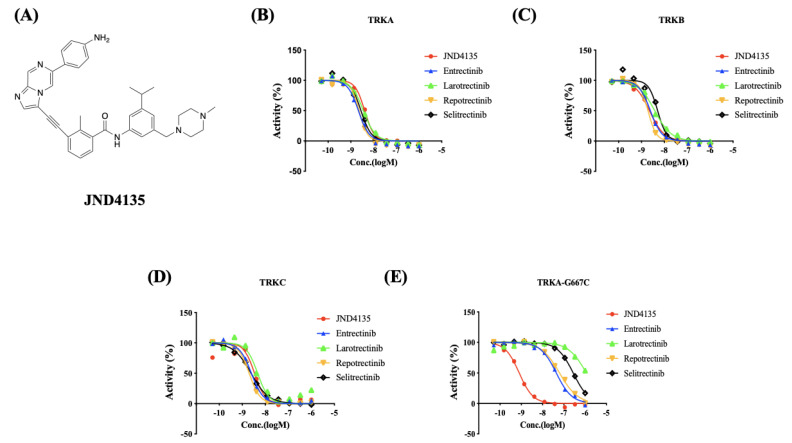
JND4135 displays TRK kinase inhibitory activities against both wild-type (WT) and resistant mutation. (**A**) structure of JND4135; and representative graphs of kinase inhibitory activities of JND4135 against TRKA (**B**), TRKB (**C**), TRKC (**D**) and TRKA-G667C (**E**). All kinase inhibitory assays included at least two independent experiments.

**Figure 2 molecules-27-06500-f002:**
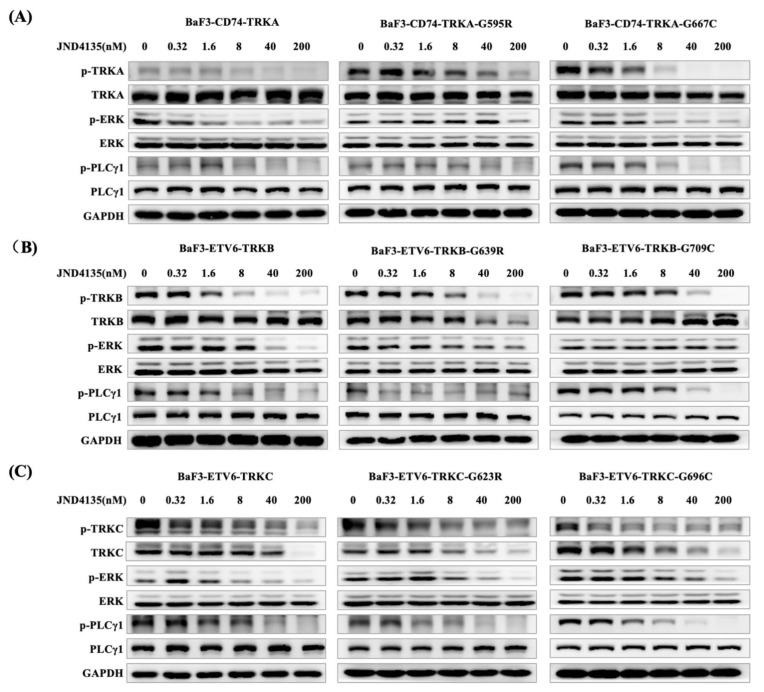
JND4135 suppresses TRK signaling pathway in TRK fusion-driven BaF3 cell lines. (**A**) JND4135 inhibited the TRKA signaling pathways in BaF3-CD74-TRKA, BaF3-CD74-TRKA-G595R and BaF3-CD74-TRKA-G667C cells. (**B**) JND4135 inhibited the TRKB signaling pathways in BaF3-ETV6-TRKB, BaF3-ETV6-TRKB-G639R and BaF3-ETV6-TRKB-G709C cells. (**C**) JND4135 inhibited the TRKC signaling pathways in BaF3-ETV6-TRKC, BaF3-ETV6-TRKC-G623R and BaF3-ETV6-TRKC-G696C cells. Cells were pretreated with 0–200 nmol/L of JND4135 for 6 h, then harvested and lysed. Cell lysates were separated by sodium dodecyl sulfate-polyacrylamide gel electrophoresis (SDS-PAGE) and analyzed by Western blot for p-TRK, p-PLCγ1 and p-Erk. GAPDH was used as a loading control.

**Figure 3 molecules-27-06500-f003:**
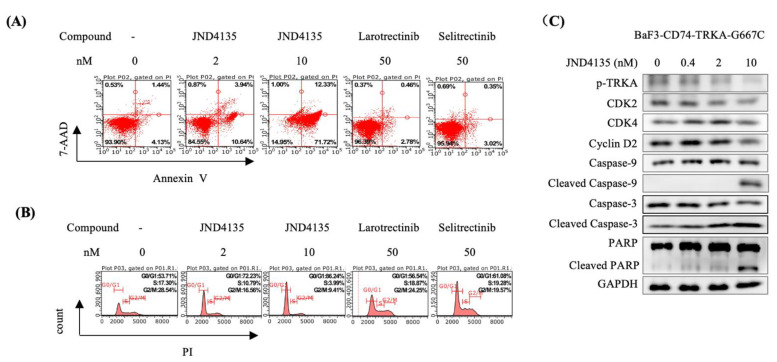
JND4135 induces apoptosis and G0/G1 cell cycle phase arrest in BaF3-TRKA-G667C cells. (**A**) JND4135 induces apoptosis in BaF3-CD74-TRKA-G667C cells. Cells were treated with indicated compounds for 48 h before Annexin V and 7-AAD co-staining. (**B**) JND4135 induces G0/G1 cell cycle arrest in BaF3-CD74-TRKA-G667C cells. Cells were treated with indicated compounds for 24 h before DNA labeling by propidium iodide. (**C**) The effect of JND4135 on the expression of p-TRKA, CDK2, CDK4, Cyclin D2, Caspase-9, Caspase-3 and PARP was tested by Western blotting. The cells were treated with JND4135 from 0.4 nM to 10 nM for 48 h. Immunoblot data are representative of three independent experiments.

**Figure 4 molecules-27-06500-f004:**
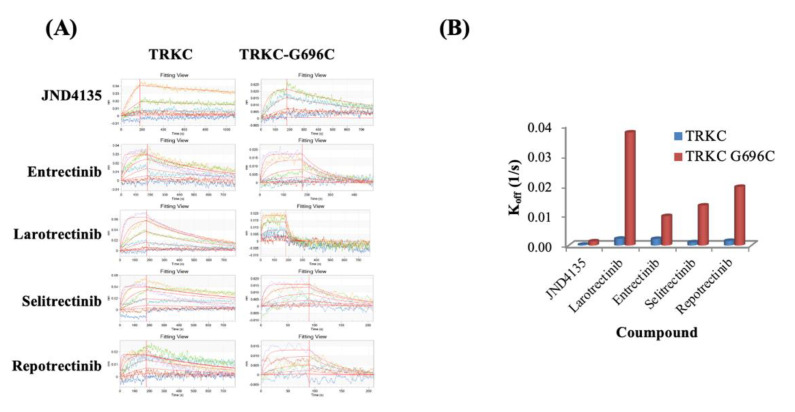
JND4135 occupies TRK protein for a long time. (**A**) Affinity activity of JND 4135 and reported TRKs inhibitors on TRKC and TRKC-G696C protein using BLI assays; (**B**) K_off_ values of JND4135 and reported TRKs inhibitors on TRKC and TRKC-G696C.

**Figure 5 molecules-27-06500-f005:**
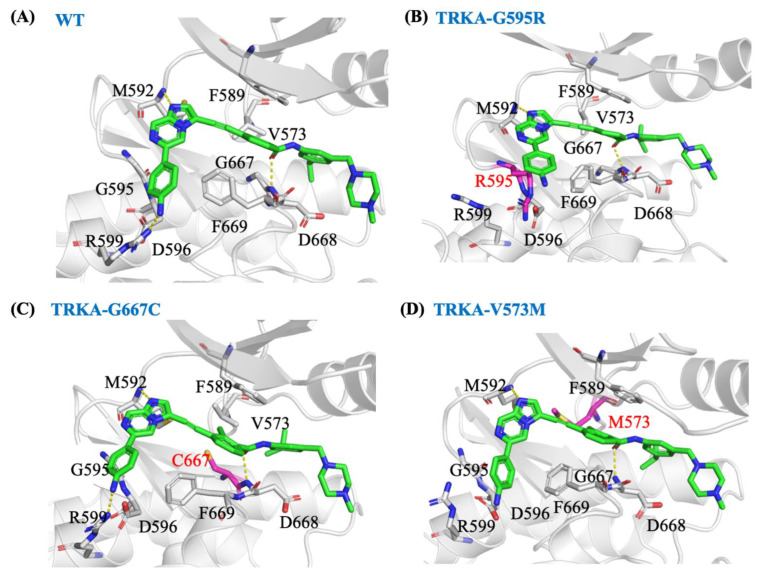
Representative structures for the binding mode of JND4135 in TRKA wild-type and mutant from 500 ns MD simulations. Predicted model of JND4135 in complex with wild type TRKA (**A**), TRKA(G595R) (**B**), TRKA(G667C) (**C**), and TRKA(V573M) (**D**). The mutant residues are shown in magenta.

**Figure 6 molecules-27-06500-f006:**
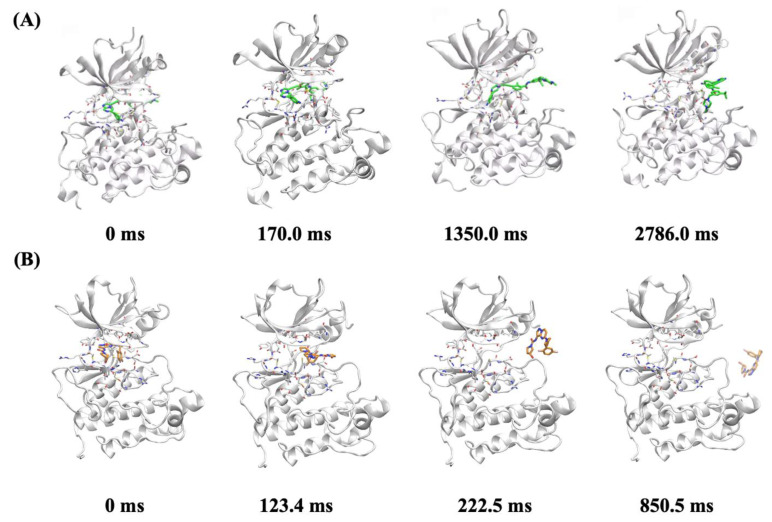
The representative structures for the unbinding process of (**A**) JND4135 from TRKA-G667C; and (**B**) larotrectinib from TRKA-G667C.

**Figure 7 molecules-27-06500-f007:**
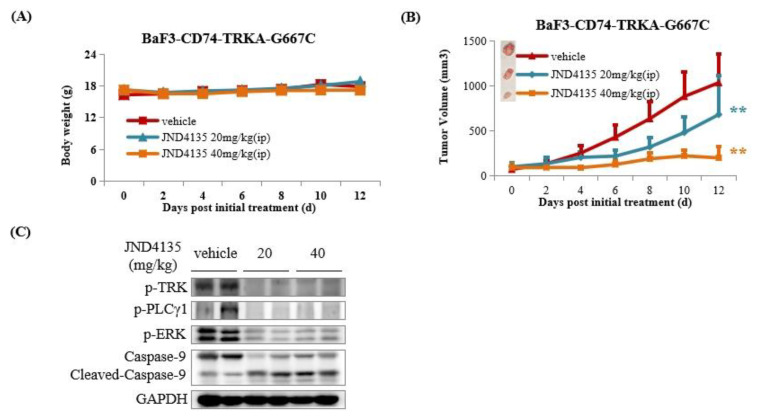
JND4135 overcomes BaF3-CD74-TRKA-G667C mutant resistance in vivo. (**A**) Antitumor efficacy of JND4135 in BaF3-CD74-TRKA-G667C xenograft mouse model. (**B**) Effect of JND4135 on body weight of BaF3-CD74-TRKA-G667C xenograft mouse model. (**C**) Effect of JND4135 on TRK signaling pathway in tumor tissues. Mice were intraperitoneally administrated JND4135 (20 and 40 mg/kg) once daily for 12 consecutive days. Tumors and body weight were measured every two days (** *p* <0.01).

**Table 1 molecules-27-06500-t001:** JND4135 overcomes a variety of TRK mutations in BaF3 cell proliferation assays. The anti-proliferative activities of the compounds were evaluated using CCK-8 assay. The data were means from at least three independent experiments.

IC50(nM ± SD)		JND4135	Entrectinib	Larotrectinib	Selitrectinib	Repotrectinib
Parental	BaF3 (+IL3)	1946 ± 25.1	2344 ± 202	>10,000	>10,000	906 ± 69.0
WT	MPRIP-TRKA	7.1 ± 2.3	1.6 ± 0.4	4.8 ± 1.2	1.1 ± 0.4	1.1 ± 0.2
	CD74-TRKA	10.4 ± 3.2	4.4 ± 1.0	8.0 ± 0.8	2.5 ± 0.2	1.6 ± 0.3
	QKI-TRKB	2.4 ± 0.6	4.8 ± 1.6	11.4 ± 4.2	1.5 ± 0.4	0.8 ± 0.7
	ETV6-TRKB	23.1 ± 5.9	8.0 ± 3.1	24.0 ± 4.2	4.9 ± 1.4	2.3 ± 0.5
	EML4-TRKC	13.2 ± 2.6	5.2 ± 1.6	21.7 ± 6.9	2.1 ± 0.2	2.7 ± 1.6
	ETV6-TRKC	10.4 ± 4.0	7.8 ± 4.2	11.2 ± 2.8	1.7 ± 1.1	2.1 ± 0.9
Solvent Front	CD74-TRKA-G595R	33.7 ± 7.1	321 ± 10.7	1110 ± 251	13.6 ± 3.3	12.0 ± 3.9
	ETV6-TRKB-G639R	10.1 ± 3.9	529 ± 130	1896 ± 778	11.4 ± 2.2	9.7 ± 1.2
	ETV6-TRKC-G623R	9.4 ± 2.6	617 ± 69.8	1269 ± 165	9.6 ± 1.2	14.6 ± 4.7
ATP site	CD74-TRKA-V573M	9.0 ± 4.8	114 ± 50.8	609 ± 195	26.2 ± 5.5	29.4 ± 1.3
	ETV6-TRKB-V617M	3.9 ± 2.6	101 ± 4.8	1963 ± 203	45.9 ± 6.3	38.4 ± 4.4
	ETV6-TRKC-V601M	11.8 ± 5.7	125 ± 56.1	2983 ± 184	52.4 ± 36.9	65.6 ± 28.9
Gatekeeper	CD74-TRKA-F589L	75.0 ± 12.7	2.1 ± 0.3	423 ± 92.6	17.3 ± 2.2	1.9 ± 0.1
	ETV6-TRKB-F633L	178 ± 4.3	13.0 ± 3.6	1396 ± 214	55.8 ± 10.8	4.9 ± 2.2
	ETV6-TRKC-F617L	37.3 ± 14.0	4.6 ± 2.8	513 ± 156	20.8 ± 7.9	2.1 ± 1.1
xDFG	CD74-TRKA-G667C	1.4 ± 0.2	33.2 ± 3.2	313 ± 48.5	66.3 ± 13	27.4 ± 5.6
	ETV6-TRKB-G709C	18.7 ± 2.9	285 ± 92.4	1628 ± 821	168 ± 73.0	89.7 ± 38.8
	ETV6-TRKC-G696C	1.3 ± 0.7	124 ± 60.4	769 ± 325	98.4 ± 55.0	42.9 ± 33.6

The anti-proliferative activities of the compounds were evaluated using CCK-8 assay. The data were means from at least three independent experiments.

## Data Availability

Data is contained within the article.

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
