# Peer review of "JND4135, a New Type II TRK Inhibitor, Overcomes TRK xDFG and Other Mutation Resistance In Vitro and In Vivo"

_molecules, 2022, doi:10.3390/molecules27196500_

Round 1

Reviewer 1 Report

This manuscript investigates the preclinical activity of JND4135, a pan-TRK inhibitor. Different methods were used in the studies, and the tumour growth inhibition activity of the compound in xenografted mice is promising. The data show that the compound could be a lead candidate for further development. It is well-written, easier to understand and with a few language errors. Overall, the work is within the scope of the journal for publication. However, there are a few major issues that need to be addressed.

# 1

Section 3.1 presents IC50 values of compounds from the kinase assay. IC50 values in the text (page 6) are in the nanomolar unit, whereas those below each graph in Figure 1B-E are in molar units. However, there is some mismatch in the presentation of data in the text. For example, the IC50 values of JND4135 against TRKA and TRKB are mentioned in the text as 2.79 and 3.19, respectively. The corresponding numbers in the molar unit are 6.966E-9 and 6.018E-9 (Figures 1B and C). How did 6.966E-9 and 6.018E-9 M become 2.79 and 3.19 nM?

Figure 1: Are the graphs shown in panel B-E representative graphs? The n number should be mentioned.

Table S1: From the reviewer's experience, the Z' Lyte Kit results in much variability between replicates. Therefore, the individual IC50s from each replicate should be shown, which were used for calculating the mean ± SD. Additionally, individual graphs should be presented in the supplementary file.

# 2

Section 3.2: IC50s of compounds in different cell lines are presented in the text, but these do not match those shown in Table 1. The IC50s in Table 1 are in micromolar units, whereas those in the text are in nanomolar, making it harder to compare. The IC50s in Table 1 should be presented in the nanomolar unit as in the text (page 6) or vice-versa.

# 3

Section 3.3: When concentration-specific statements are made on the effects of drugs, they should be verified statistically. For example, on page 6 (line 296), it is mentioned that 1.6 nM JND4135 decreased TRKA-Wt levels, but the blot (Fig 2A, BaF3-CD74-TRKA) does not support the statement. Similarly, the effect of JND4135 on TRKB-G639R at 8 nM (page 7, line 304) is not shown in the blot (it appears the effect is more prominent at 40 nM).

Figure S1: It was not cited in the manuscript. Additionally, the ERK bands increase with an increase in compound concentration, which is opposite to that in Fig 2 in transfected cell lines. This should be explained.

Figure S2 shows a decrease in TRK expression after treatment with JND and other compounds. This should be mentioned.

# 4

Section 3.4: Some of the blots shown in Fig 3C are not convincing. Line 313-316: There is a reference to cyclin E1, but no cyclin E1 blot exists in Figure 3C. 

The proof that JND cleaved caspase 3 is not strong. A better representative blot should be shown.

Caspase-3 is known to cleave PARP-1. What is the evidence that PARP-1 cleavage seen in Fig 3C is not because of the compound but rather caspase 3?

There is a mismatch in the labelling of Fig A and B. The legend says Fig A shows propidium iodide plots, but these are annexin plots.

In the legend of Fig 3B, it is mentioned that cells were treated with 2, 10 and 50 nM of compounds for 48 before DNA labelling by Annexin V and 7-AAD. The sentence should be corrected because it reads as though Annexin V was used for DNA labelling.

On line 316, it is mentioned, '…in both cell lines (Fig 3C)'….Which are both cell lines? Fig 3 shows blots from only a single cell line.

# 5

Section 3.5: Line 328: '……significantly different kinetics parameters of binding….' How was this significance determined? There are no statistical comparisons in the Tables.

# 6

Page 9, line 421: For in vivo studies, xenografted mice were used. For ADME studies, did the authors use rats? Please clarify.

It should be Table S3, not Table S3 (Line 421).

# 7

Figure 7C (legend): it should be 12 consecutive days, not 13. 

Author Response

Reviewer: 1

Comments:

This manuscript investigates the preclinical activity of JND4135, a pan-TRK inhibitor. Different methods were used in the studies, and the tumour growth inhibition activity of the compound in xenografted mice is promising. The data show that the compound could be a lead candidate for further development. It is well-written, easier to understand and with a few language errors. Overall, the work is within the scope of the journal for publication. However, there are a few major issues that need to be addressed.

First of all, thank you very much for your recognition and affirmation of our work

Question 1: Section 3.1 presents IC+ values of compounds from the kinase assay. IC50 values in the text (page 6) are in the nanomolar unit, whereas those below each graph in Figure 1B-E are in molar units. However, there is some mismatch in the presentation of data in the text. For example, the IC50 values of JND4135 against TRKA and TRKB are mentioned in the text as 2.79 and 3.19, respectively. The corresponding numbers in the molar unit are 6.966E-9 and 6.018E-9 (Figures 1B and C). How did 6.966E-9 and 6.018E-9 M become 2.79 and 3.19 nM?

Figure 1: Are the graphs shown in panel B-E representative graphs? The nnumber should be mentioned.

Table S1: From the reviewer's experience, the Z' Lyte Kit results in much variability between replicates. Therefore, the individual IC50s from each replicate should be shown, which were used for calculating the mean ± SD. Additionally, individual graphs should be presented in the supplementary file.

RESPONSE: 

Thanks very much for your comments and suggestion!

We apologized for the mismatch between the graphs and text. Figure 1.B-E are representative graphs from single experiment results, however the data of text are mean±SD from at least three repeated results. To avoid misunderstanding, we have deleted IC50 values in Figure 1.B-E and added the statement as:representative graphs of kinase inhibitory activities of JND4135 against TRKA (B), TRKB (C),TRKC (D) and TRKA-G667C (E). All kinase inhibitory assays performed at least two independent experiments.  

For Table S1, we have added the individual IC50s and graphs from each replicate in Supplementary File.

Question 2: Section 3.2: IC50s of compounds in different cell lines are presented in the text, but these do not match those shown in Table 1. The IC50s in Table 1 are in micromolar units, whereas those in the text are in nanomolar, making it harder to compare. The IC50s in Table 1 should be presented in the nanomolar unit as in the text (page 6) or vice-versa.

RESPONSE: Thanks very much for your comments! This is really a good suggestion! We have already changed the numbers into nanomolar unit in the Table 1 and Table S3.

Question 3: Section 3.3: When concentration-specific statements are made on the effects of drugs, they should be verified statistically. For example, on page 6 (line 296), it is mentioned that 1.6 nM JND4135 decreased TRKA-Wt levels, but the blot (Fig 2A, BaF3-CD74-TRKA) does not support the statement. Similarly, the effect of JND4135 on TRKB-G639R at 8 nM (page 7, line 304) is not shown in the blot (it appears the effect is more prominent at 40 nM).

Figure S1: It was not cited in the manuscript. Additionally, the ERK bands increase with an increase in compound concentration, which is opposite to that in Fig 2 in transfected cell lines. This should be explained.

Figure S2 shows a decrease in TRK expression after treatment with JND and other compounds. This should be mentioned.

RESPONSE:

Thanks very much for your comments and suggestion! We have already corrected the statements according to more prominent concentration ranges as following: JND4135 decreased the  phosphorylation levels of TRKA and its downstream signaling proteins ERK, PLCγ from 8 nM for TRKA-WT, 40 nM for TRKA -G595R and 1.6 nM for TRKA-G667C cells.”

For figure S1, we have added the statement and explanation in revised manuscript as following: “We also investigated the effects of JND4135 on cellular signal pathway alterations in parental BaF3 cells (Fig. S1), which were lack of TRK fusion proteins and driven by interleukin-3 (IL-3). It was shown that JND4135 treatment increased the phospho-ERK levels, which probably resulted from negative feedback after JND4135 treatment in parental BaF3 cells.”

For Figure S2, we have added the statement in revised manuscript as following: Interestingly, it was noted that TRK total protein decreased at high concentration JND4135 treatment in some BaF3-TRKs cells, especially the TRKC fusion proteins (Fig 2 & Fig. S2), which were consistent with some reported data of TRK inhibitors by other groups (17,33-35)”

Question 4:

Section 3.4: Some of the blots shown in Fig 3C are not convincing. Line 313-316: There is a reference to cyclin E1, but no cyclin E1 blot exists in Figure 3C.

The proof that JND cleaved caspase 3 is not strong. A better representative blot should be shown.

Caspase-3 is known to cleave PARP-1. What is the evidence that PARP-1 cleavage seen in Fig 3C is not because of the compound but rather caspase 3?

There is a mismatch in the labelling of Fig A and B. The legend says Fig A shows propidium iodide plots, but these are annexin plots.

In the legend of Fig 3B, it is mentioned that cells were treated with 2, 10 and 50 nM of compounds for 48 before DNA labelling by Annexin V and 7-AAD. The sentence should be corrected because it reads as though Annexin V was used for DNA labelling.

On line 316, it is mentioned, '…in both cell lines (Fig 3C)'….Which are both cell lines? Fig 3 shows blots from only a single cell line.

RESPONSE: Thanks very much for your comments and suggestions!

We apologized for the mistakes. We have replaced the bands of caspase-3 and cleaved caspase-3 with better ones, and corrected mistakes in legends as following:

We agreed your opinion that PARP is cleaved by Caspase 3 in apoptosis. We have deleted “cyclin E1” in text and revised related sentences as following:  WB results further validate that JND4135 dose-dependently diminished the cell cycle related protein levels of cyclin-dependent kinase 2 (CDK2), Cyclin D2and activated the caspase apoptotic cascade resulting in the cleavage of PAPR in BaF3–CD74-TRKA-G667C cells.” Thank you again for your helpful suggestions.

Question 5:

Section 3.5: Line 328: '……significantly different kinetics parameters of binding….' How was this significance determined? There are no statistical comparisons in the Tables.

RESPONSE: Thanks very much for your kindly comments! We have revised this section as following: “Interestingly, we also found that the dissociation rates of JND4135 with TRKC-WT and TRKC-G696C were 2.78E-4 1/s and 1.35E-3 1/s, respectively, which were approximately 3.81-27.85 folds slower than that of control drugs (Table S2, Fig.4B). Therefore, once binding with TRKs protein, JND4135 could occupy the protein for a longer time than the control drugs.”

Question 6: 

Page 9, line 421: For in vivo studies, xenografted mice were used. For ADME studies, did the authors use rats? Please clarify.

It should be Table S3, not Table S3 (Line 421).

RESPONSE: Thanks very much for your kindly comments and suggestions! We used rats for ADME studies. We have already revised these errors as following: “However, JND4135 displayed almost no absorption via oral administration in rats (Table S4).”

Question 7: 

Figure 7C (legend): it should be 12 consecutive days, not 13.

RESPONSE: Thanks very much for your kindly suggestion! We have already revised these errors.

Reviewer 2 Report

In this manuscript, Wang, et al. reported JND4135, a new type II pan-TRK inhibitor, could overcome TRK mutant resistance, including the xDFG mutations in vitro and in vivo. This work is of significance for related research filed. However, the manuscript has some problems which should be addressed before acceptance.

1.     The compound name should be consistent throughout the manuscript. For example, the names of positive control such as Larotrectinib (LOXO-101), Entrectinib (RXDX-101), Selitrectinib (LOXO-195) and Repotrectinib (TPX-0005) in some figures are not consistent with that in the Results section. Another one is “JND4135” and “D4135” in Figure S1 and Table S4.

2.     Figure 1, EC50 should be changed to IC50.

3.     A very serious problem is that the inhibitory activities of these compounds including JND4135 and positive control drugs in Figure 1 are different from the text described in the Results section. The authors should give a reasonable explanation.

4.     In Figure 2, there was no western blot analysis for p-AKT and AKT, which was described in the legends of Figure 2. Please supplement corresponding blots. In addition, the catalog numbers for AKT and p-AKT antibodies should be given in the Materials and Methods.

5.     Figure S1 was not mentioned in the manuscript, please correct.

6.     The English spelling and grammar should be carefully checked.

Author Response

Reviewer 2

Comments:

In this manuscript, Wang, et al. reported JND4135, a new type II pan-TRK inhibitor, could overcome TRK mutant resistance, including the xDFG mutations in vitro and in vivo. This work is of significance for related research filed. However, the manuscript has some problems which should be addressed before acceptance.

First of all, thank you very much for your recognition and affirmation of our work

Question 1:

The compound name should be consistent throughout the manuscript. For example, the names of positive control such as Larotrectinib (LOXO-101), Entrectinib (RXDX-101), Selitrectinib (LOXO-195) and Repotrectinib (TPX-0005) in some figures are not consistent with that in the Results section. Another one is “JND4135” and “D4135” in Figure S1 and Table S4.

RESPONSE: Thanks very much for your kindly suggestion! We have already revised these errors.

Question 2:

Figure 1, EC50 should be changed to IC50.

RESPONSE: Thanks very much for your comments and suggestion! We have revised these errors.

Question 3:

A very serious problem is that the inhibitory activities of these compounds including JND4135 and positive control drugs in Figure 1 are different from the text described in the Results section. The authors should give a reasonable explanation.

RESPONSE:

Thanks very much for your comments and suggestion!

We apologized for the mismatch between the graphs and text. Figure 1.B-E are representative graphs from single experiment results, however the data of text are mean±SD from at least three repeated results. To avoid misunderstanding, we have deleted IC50 values in Figure 1.B-E and added the statement as: “representative graphs of kinase inhibitory activities of JND4135 against TRKA (B), TRKB (C), TRKC (D) and TRKA-G667C (E). All kinase inhibitory assays performed at least two independent experiments”.  

Question 4:

In Figure 2, there was no western blot analysis for p-AKT and AKT, which was described in the legends of Figure 2. Please supplement corresponding blots. In addition, the catalog numbers for AKT and p-AKT antibodies should be given in the Materials and Methods.

RESPONSE:

Thanks very much for your comments and suggestion!

We are sorry for the written errors. We have not detected the pAKT in Figure 2 because it had reported that repotrectinib cannot effectively inhibit the p-AKT levels in TRK driven cancer cells (Mol Cancer Ther. 2021,20(12):2446-2456.).

We have deleted the pAKT in legends of Figure 2 in revised manuscript.

Question 5:

Figure S1 was not mentioned in the manuscript, please correct.

RESPONSE: Thanks very much for your comments and suggestion! We have added some statements regarding Figure S1 in the revised manuscript as following: “We also investigated the effects of JND4135 on cellular signal pathway alterations in parental BaF3 cells (Fig. S1)…”

Question 6:

The English spelling and grammar should be carefully checked.

RESPONSE: Thanks very much for your comments and suggestion! We have already carefully revised these spelling, grammar and format errors.